# Vitamin D and Chronic Diseases among First-Generation Immigrants: A Large-Scale Study Using Canadian Health Measures Survey (CHMS) Data

**DOI:** 10.3390/nu14091760

**Published:** 2022-04-22

**Authors:** Said Yousef, Ian Colman, Manny Papadimitropoulos, Douglas Manuel, Alomgir Hossain, MoezAlIslam Faris, George A. Wells

**Affiliations:** 1Faculty of Medicine, School of Epidemiology and Public Health, University of Ottawa, Ottawa, ON K1G 5Z3, Canada; icolman@uottawa.ca (I.C.); alhossain@ottawaheart.ca (A.H.); gawells@ottawaheart.ca (G.A.W.); 2Cardiovascular Research Methods Centre, University of Ottawa Heart Institute, Ottawa, ON K1Y 4W7, Canada; 3Eli Lilly Canada Inc., Toronto, ON M5X 1B1, Canada; papadimitropoulos_manny@lilly.com; 4Faculty of Pharmacy, University of Toronto, Toronto, ON M5S 3M2, Canada; 5Ottawa Hospital Research Institute, Ottawa, ON K1Y 4E9, Canada; dmanuel@ohri.ca; 6Institute for Clinical Evaluative Sciences, Ottawa, ON K1Y 4E9, Canada; 7Department of Family Medicine, University of Ottawa, Ottawa, ON K1H 8M5, Canada; 8Department of Clinical Nutrition and Dietetics, Research Institute of Medical and Health Sciences (RIMHS), College of Health Sciences, University of Sharjah, Sharjah 27272, United Arab Emirates; mfaris@sharjah.ac.ae

**Keywords:** immigrants’ health deterioration, vitamin D, serum 25-hydroxyvitamin D, chronic diseases

## Abstract

**Background:** Nearly 22% of the Canadian population are first-generation immigrants. We investigated immigrants’ health status and health deterioration over time in terms of the prevalence of chronic diseases (CDs) and their relationship to vitD status. **Methods:** We used cycles three (2012–2013) and four (2014–2015) of the Canadian Health Measures Survey. These data contained unique health information and direct physical/blood measures, including serum 25-hydroxyvitamin D (S-25(OH)D). Indicators of health status and deterioration were the prevalence of CDs diagnosed by healthcare professionals, self-reported general and mental health, and CD-related biomarkers. **Results:** The data (*n* = 11,579) included immigrants from more than 153 countries. Immigrants were healthier than non-immigrants for most health status measures. The prevalence of CDs was higher among those who migrated to Canada aged ≥ 18 years. A longer time in Canada after immigration was associated with a higher risk for CDs. The mean S-25(OH)D was lower among immigrants, higher among patients with CDs, and inversely associated with glycated hemoglobin, total cholesterol/high-density lipoprotein ratio, immunoglobulin E, serum ferritin, and blood hemoglobin. After adjusting for covariates, no association was found between S-25(OH)D and the prevalence of CDs. **Conclusions:** Lower levels of accumulated S-25(OH)D among immigrants may impact their health profile in terms of CD-related biomarkers, which partially explains immigrants’ health deterioration over time. We recommend further longitudinal research to investigate immigrants’ vitD and health deterioration.

## 1. Introduction

Immigrants are a fundamental part of the Canadian population and policy framework [1], with nearly 22% of the Canadian population being first-generation immigrants [2]. Statistics Canada estimated an increase of approximately 1 million foreign-born residents every 3 years by 2035/2036 [3]. Therefore, the health of immigrants will impact Canada’s overall future healthcare system.

Accumulated evidence suggests that recent immigrants have better health than people of the host country, including long-term immigrants who had previously migrated to the host country. This phenomenon is called the “healthy immigrant effect” or “healthy immigrant advantage” [4,5,6]. However, immigrants’ physical and mental health declines after arrival in Canada, with health changes often occurring within 5–10 years [4,5]. Pre-migration factors, such as rigorous selection criteria that favor immigrants that are healthier, wealthier, and better educated, along with post-immigration factors, such as environmental and lifestyle changes, may explain this health decline after migration [4,7,8,9,10]. In addition, the healthy immigrant effect varies through the course of life [6].

Moreover, more immigrants have deficient and insufficient vitamin D (vitD) levels compared with native-born populations [2,11,12]. However, vitD status varies among immigrants because of ethnic variations, skin pigmentation, resettlement changes in diet, physical activity, and sun exposure [2,11,13,14]. The concentration of serum 25-hydroxyvitamin D (S-25(OH)D) represents the combined contributions of the cutaneous synthesis and dietary intake of vitD and is considered the most important clinical marker for the overall vitD level [12,15,16]. A lower concentration of S-25(OH)D was found to be a risk factor for CDs [17,18,19,20] and could predict all-cause mortality [17,18,21]. Moreover, Jablonski et al. expected the diseases associated with vitD deficiency to be a major source of morbidity and mortality in the 21st century [22]. Most recent guidelines have emphasized the importance of vitD for skeletal health, although evidence supporting its benefits for non-skeletal health outcomes is either weak or conflicting [23,24]. There is a range of recommended vitD thresholds that are used to study health effects. Mean S-25(OH)D (nmol/L) values or ranges at various thresholds (e.g., deficiency: 25–30 nmol/L; insufficiency: 25–49 nmol/L; and sufficiency: 50–75 nmol/L) are commonly used in studies to describe vitD status [12]. An insufficient vitD level (<50 nmol/L) is more frequently used to describe hypovitaminosis D [25].

vitD deficiency has been linked to physical and mental illnesses [26,27]. Recent research supported the hypothesis that vitD deficiency was related to the incidence, severity [28,29], and mortality attributed to COVID-19 [29,30]. These studies concluded that those at the highest risk for severe COVID-19 matched those at the highest risk for severe vitD deficiency, including older people, darker-skinned ethnic groups, and obese people [28,30,31]. Moreover, research has highlighted the importance of further studies investigating immigrants’ health deterioration in the context of vitD [2,32], lifelong adversity [33], and changes in lifestyle and dietary intake [5,32]. However, leading causes of death, such as CDs (cardiovascular disease, cancer, chronic lung disease, and diabetes), are used as reliable measures for public health [34].

The main objective of this study was to investigate immigrants’ health status and possible health deterioration in relation to serum vitD levels. Moreover, we used CDs and CD-related biomarkers as proxy indicators to investigate risk factors for health deterioration in the context of vitD status. In addition, we compared the health status (in relation to the vitD status) of immigrants from different ethnic groups and origins with non-immigrants.

## 2. Materials and Methods

### 2.1. Study Design and Participants

We previously published the main characteristics, mean S-25(OH)D levels, and prevalence and leading determinants of vitD deficiency/insufficiency among immigrants and non-immigrants in Canada [2]. The present study followed the Strengthening the Reporting of Observational Studies in Epidemiology (STROBE) statement in the planning, implementation, and reporting [35].

Health and vitD status among first-generation immigrants (foreign-born) were ascertained using Canadian Health Measures Survey (CHMS) data and compared with those in the non-immigrant (native-born) population. The CHMS represents the first national data on vitD, includes the Canadian immigrant population, addresses gaps in existing national information, and contains unique health information, including direct physical and blood measures [36,37]. The sample size for each CHMS cycle was carefully selected to provide a reliable and representative estimate at the national level for sex and age groups. The survey covered approximately 96% of the Canadian population. A dwelling stratification stage was applied, followed by a roster list of all people living in each household, from which individuals aged 3–79 years were randomly selected [36,37]. The CHMS sample population weight was adjusted for age and sex across Canada’s five standard geographic regions. All participants provided written informed consent, and the CHMS was approved by the Health Canada Research Ethics Board [36,37].

### 2.2. Measures

The S-25(OH)D is expressed in nanomoles per liter (nmol/L) and is measured using chemiluminescence immunoassay technology (DiaSorin^®^, Ltd., Stillwater, MN, USA). The analytical detection limit for S-25(OH)D was 10–375 nmol/L. We used S-25(OH)D cut-off points to identify vitD status as either sufficient (≥50 nmol/L) or insufficient (<50 nmol/L), as defined by the Institute of Medicine (IOM) and other experts [12,38,39].

We used the presence of CDs as the outcome-dependent factor for health deterioration. CDs included 24 chronic diseases and conditions (e.g., type 1 and 2 diabetes, heart disease (not limited to ischemic heart disease), and mood disorder). Respondents were asked “Do you have a long-term condition that was diagnosed by a healthcare professional and expected to last or has already lasted 6 months or more?” (yes/no). We used CDs (e.g., diabetes, heart disease, cancer, and asthma) for which a date of diagnosis was listed. We combined CDs that had similar inflammatory characteristics (including, but not limited to chronic obstructive pulmonary disease, bronchitis, and emphysema) as one variable called “inflammatory lung diseases.” Inflammatory lung diseases were measured for participants aged > 29 years, but no date of diagnosis was available. CDs were considered present if at least one of the five listed health conditions (diabetes, heart disease, cancer, asthma, and inflammatory lung diseases) received a “yes” response as a separate variable called “CDs”. The date of diagnosis for the CDs was not available.

Our analysis examined: five health conditions (diabetes, heart disease, cancer, asthma, and inflammatory lung diseases), CDs (at least one), self-rated general health, and self-rated mental health. We also considered seven biomarkers associated with CDs: glycated hemoglobin (HbA1c), the total cholesterol/high-density lipoprotein cholesterol ratio (TC/HDL ratio), high-sensitivity C-reactive protein (hs-CRP), immunoglobulin E (IgE), ferritin, hemoglobin (Hb), and S-25(OH)D (the serum concentration of vitD). In addition, 16 sociodemographic and health behavior variables were examined as independent factors: immigration status, sex, age, income, education level, body mass index (BMI; kg/m^2^), smoking status, alcohol consumption, age at immigration, length of time in Canada, physical activity, region, country of birth, ethnicity, dairy intake, and vitD supplement use.

The CHMS was a cross-sectional survey. Data for health measures, including several temporal attributes such as date of immigration and mobile clinic visit dates (e.g., blood sampling including S-25(OH)D), were used to create proxy health indicators for health deterioration. To examine longitudinal changes in health, we created a new variable to compare “diagnosis before immigration” versus “diagnosis after immigration”. We also created a new variable “time after diagnosis” to investigate the relationship between S-25(OH)D and the presence of CDs. We used two categories (recent vs. long-term) for diagnosis at two time-points (1 year and 5 years). We used the lowest cut-off value to represent the most recent S-25(OH)D measures at the date of diagnosis (e.g., ≤1 year and ≤5 years) and the highest to represent the long-term measures for S-25(OH)D at the date of diagnosis (e.g., >1 year and >5 years).

A Likert scale was used for the two self-reported questions that reflected the quality of general and mental health. Participants aged over 12 years rated these questions, with the ratings for each question dichotomized in two categories: “fair/poor” versus “good” (excellent/very good/good).

Detailed information about the CHMS data, methods, and merging of the two cycles can be found in our previous work [2] and on the Statistics Canada website (http://www.statcan.gc.ca, accessed 14 February 2021) [40,41]. The method of measurement and the analytical detection limit for each biomarker are available in Appendix A, Table A1.

### 2.3. Statistical Analysis

Following the recommendations of Statistics Canada, all analyses used population weights that reflected the probability a respondent was included in the survey. Descriptive statistics were used to investigate immigrants’ demographic, clinical, and behavioral characteristics compared with non-immigrants. We used the mean and standard error (SE) for continuous variables. Data were stratified based on the study outcomes (CDs) and variables of interest (e.g., immigration status). To account for the unequal probability of selection and present an accurate estimate of the Canadian population, we used the survey command, recommended sample weight, and degrees of freedom in the analyses. Therefore, all results were weighted values. In addition to the two S-25(OH)D cutoff points (sufficient ≥ 50 nmol/L and insufficient < 50 nmol/L), we used the continuous values for S-25(OH)D and other biomarkers.

We used univariate analyses to identify independent covariates. Multivariable logistic regression models (odds ratio (OR) and 95% confidence interval (CI)) were then used to evaluate the association between CDs (indication of health deterioration) and vitD status for immigrants compared with non-immigrants. The final model was adjusted for immigration status, age, sex, household income, education, ethnicity, BMI, physical activity, smoking behavior, alcohol consumption, vitD supplement use, HbA1c, the TC/HDL ratio, Hb, ferritin, IgE, and hs-CRP. Statistical significance was set at *p* ≤ 0.05. All analyses were performed with SPSS version 26.0 (IBM Corp., Armonk, NY, USA) and Stata version 16.0 (StataCorp, College Station, TX, USA).

## 3. Results

We examined data for 11,579 CHMS participants, including immigrants from more than 153 countries. The main characteristics of the study sample have previously been published [2]. The prevalence rates of CDs based on the sociodemographic and immigration characteristics of all study participants are presented in Table 1. Immigrants had a lower prevalence of CDs than non-immigrants. Those who had migrated to Canada at age ≥ 18 years had a higher prevalence of CDs than those who migrated at under 18 years of age. At two different time points after the migration (5 and 10 years), the prevalence was higher among those who had stayed in Canada for ≥5 years and ≥10 years than among those who had stayed less than 5/10 years. The prevalence of CDs was higher among immigrants with older age, lower household income, underweight, overweight, and obesity compared with their younger, higher income, and average body weight counterparts. Participants who did not meet the Canadian physical activity recommendations had a higher prevalence of CDs.

A lower prevalence of CDs was observed among Chinese, Blacks, Filipinos, Arabs, and the “all other ethnicities” group compared with the white group (Table A2). The prevalence of CDs based on region and place of birth is presented in Table A3. Those born in South/Central America, the Caribbean, Africa, and Asia had a lower prevalence of CDs than those born in Canada and North America. A detailed analysis based on the country of birth showed that, compared with native-born Canadians, more participants born in Italy had CDs and fewer of those born in the Philippines had CDs.

Compared with non-immigrants, immigrants had a lower prevalence of heart disease, cancer, asthma, inflammatory lung diseases, and CDs but a higher prevalence of vitD insufficiency (52.82 vs. 31.75, *p* < 0.001). Immigrants also had lower mean concentration levels of S-25(OH)D, serum hs-CRP, and Hb and higher mean levels of HbA1c, the TC/HDL ratio, and ferritin than non-immigrants (Table 2).

Immigrants with diabetes, heart disease, and poor/fair self-reported general and mental health had lower S-25(OH)D levels than non-immigrants. More immigrants with diabetes, heart disease, cancer, asthma, inflammatory lung diseases, CDs, and poor/fair general or mental health had insufficient vitD levels compared with non-immigrants. However, there was no difference in vitD supplement use for any health indicators between immigrants and non-immigrants (Table 3).

There were no differences in S-25(OH)D levels between patients diagnosed before and after immigration for these health indicators (Table 4). There were no dates of diagnosis for inflammatory lung diseases, CDs, and self-reported general and mental health.

The results shown in Appendix A and Table A4, Table A5, Table A6 and Table A7 represent all study participants and are not stratified by immigration status or CDs. No differences were observed in mean S-25(OH)D levels by recent or long-term diagnosis (Table A4). The data were not analyzed for inflammatory lung diseases, CDs, and self-reported general and mental health because the date of diagnosis was not given.

The mean values for S-25(OH)D, vitD status (sufficiency/insufficiency), and the use of vitD supplements for each health indicator, including CDs, are presented in Table A5. Patients diagnosed with diabetes had insufficient vitD compared with those without diabetes. Patients with heart disease, CDs, and who self-reported poor/fair general health had higher S-25(OH)D levels than their counterparts without heart disease or CDs and with good general health. Patients with cancer tended to use more supplements than those without cancer. In addition, the mean concentration levels of S-25(OH)D, HbA1c, and IgE were higher among patients with CDs than among those without CDs (Table A6).

The linear regression analysis showed that when the concentration levels of S-25(OH)D increased by 1 nmol/L, HbA1c, the TC/HDL ratio, IgE, ferritin, and Hb decreased (Table A7). In the multivariable logistic regression analysis, S-25(OH)D was associated with the prevalence of CDs (unadjusted OR: 1.006; 95%CI: 1.00, 1.01; *p* = 0.007) (result not shown). After adjusting for covariates (Table 5), the final model indicated there was no association between S-25(OH)D and CDs (OR: 1.007; 95%CI: 1.00, 1.01; *p* = 0.070). Immigrants were healthier (OR for CDs decreased by 60%) than non-immigrants. We also found that when age increased by 1 year, the OR for CDs increased by 2%. The OR increased by 71% among obese participants, by 174% when the HbA1c increased by 1%, and by 0.001% when IgE increased by 1 IU/mL.

## 4. Discussion

This study used data for a large population of native-born Canadians and immigrants from 153 countries. Immigrants had a lower prevalence of CDs than native-born Canadians, including heart disease, cancer, asthma, and inflammatory lung diseases. Immigrants who came to Canada aged ≥ 18 years had more CDs than those younger than 18 years. There was no difference in the prevalence of CDs before immigration compared with post immigration. However, using two different time points (5 and 10 years), we found that the longer immigrants had lived in Canada, the higher the prevalence of CDs, which may be considered an early sign of health deterioration. Although the CHMS used a cross-sectional design, we used proxy indicators to verify the existence of the healthy immigrant effect (immigrants’ health advantage in Canada).

Our results were consistent with those of Vang et al. (2015) who conducted a systematic review involving 77 studies of migration and health in Canada. That study found the healthy immigrant effect was more noticeable among recent immigrants and gradually disappeared among immigrants that were well-established in the host country. Moreover, the advantage of the healthy immigrant effect was also found in terms of protecting immigrants against CDs, including cancer, diabetes, heart disease, asthma, and obesity [6]. Those authors indicated there was a significant deviation for morbidity over time in the healthy immigrant effect, and there were no identifiable underlying reasons for health deterioration [6]. Moreover, Paszat et al. (2017) found that immigration was a determinant for developing colorectal cancer after the first 10 years of arrival in Canada [42]. Another study reported that Canadian immigrants demonstrated lower cancer-specific mortality, but this benefit diminished over time. After adjusting for age, each year following arrival was associated with increased mortality [43].

We hypothesized that lower S-25(OH)D levels, along with environmental and behavioral changes, may explain health deterioration. However, the relationship between vitD and CDs and the acculturation process may be affected by many factors, such as whether immigrants adopted healthy or unhealthy behaviors after immigration compared with before immigration. For example, in previous immigration, acculturation, and vitD studies, it was reported that immigrants less frequently consumed vitD-rich foods, engaged in less physical activity, and were less exposed to sun than non-immigrants [2,6]. Other studies found the length of residency since immigration was a crucial indicator of lifestyle acculturation. Higher acculturation levels were associated with significantly higher S-25(OH)D [2,5,44].

This study found that, compared with non-immigrants, immigrants had lower mean S-25(OH)D levels and more vitD insufficiency. Similarly, immigrant patients with CDs (e.g., diabetes, heart diseases, cancer, asthma, and inflammatory lung diseases) and poor/fair self-reported general and mental health had higher insufficiency levels than non-immigrants. In a systematic review and meta-analysis of 95 studies (880,128 participants), Chowdhury et al. (2014) reported an inverse association between circulating S-25(OH)D and the risk for death due to cardiovascular disease, cancer, and other mortality causes. Moreover, vitD3 supplementation was inversely associated with lower overall mortality among older adults [17]. Several other studies, including reviews, found that inadequate S-25(OH)D concentration was associated with an increased risk of CDs [26,27]. Despite the high level of evidence supporting these findings, there remains some controversy regarding the causative nature and efficacy of vitD. For example, there was a two-way correlation between vitD deficiency and disease outcomes such as infectious diseases, rheumatoid arthritis, and obesity [45].

Our previous findings indicated that immigrants had higher S-25(OH)D levels the longer they lived in Canada [2]. To gain better insights about the relationship between the duration of diagnoses and the prevalence of CDs and S-25(OH)D levels, we hypothesized that those who were recently (≤1 year or ≤5 years) diagnosed would have lower S-25(OH)D levels than those with a long duration of diagnosis (>1 year or >5 years). A previous study reported that S-25(OH)D may be sensitive to changes in health status [19]. However, our analysis revealed no difference in S-25(OH)D between recently diagnosed patients and their counterparts with a longer time since diagnosis.

Another study recommended investigating pre-and post-migration experiences to better understand the healthy immigrant effect and health changes over time in the host country [6]. Therefore, we evaluated the prevalence of CDs and pre-and post-immigration vitD by the date of diagnosis and date of immigration. The results showed no differences in mean S-25(OH)D among immigrants diagnosed before immigration compared with those diagnosed after immigration for any of the studied health indicators (diabetes, heart disease, cancer, and asthma).

We assumed that having lower S-25(OH)D levels (deficient or insufficient vitD) for a long time may increase the risk of a CD diagnosis. Therefore, we investigated the relationships between the length of time since immigration and vitD status and CDs. Our results showed that the longer the time after immigration, the higher the prevalence of CDs among immigrants.

Our univariate analyses showed that the mean concentrations of CD-related biomarkers (HbA1c, hs-CRP, and IgE) were higher among patients with CDs compared with those without CDs. We did not expect to find the mean concentration of S-25(OH)D was higher among patients with CDs compared with their non-CD counterparts: mean (SE) 63.118 (1.787) versus 59.42 (1.76) (95%CI: −6.27, −1.13; *p* = 0.007). However, the findings related to immigration status and the use of vitD supplements along with other covariates may change the direction of the relationship between CDs and S-25(OH)D. For example, non-immigrants had more CDs and higher S-25(OH)D, and patients with CDs used more vitD supplementation than those without CDs. After adjusting for immigration status, vitD supplementation, and other covariates, S-25(OH)D was not associated with CDs, although HbA1c and IgE remained associated with a higher prevalence of CDs.

Our analysis showed a negative association between S-25(OH)D and HbA1c, the TC/HDL ratio, IgE, ferritin, and Hb levels. Using the same data for Canadian adults, another study found the same inverse association between S-25(OH)D and elevated ferritin [46]. In addition, vitD supplementation was found to improve S-25(OH)D levels among patients with diabetes with vitD deficiency [47]. Our investigation of the concentrations of these biomarkers among immigrants compared with non-immigrants showed that immigrants had higher concentrations of HbA1c, and ferritin, a higher TC/HDL ratio, and lower levels of hs-CRP and Hb than non-immigrants. These deviations in CD-related biomarkers may be a preliminary indicator for the decline in the health status of immigrants in the long term.

Our univariate analyses revealed that immigrants with insufficient vitD were more likely to self-report poor/fair general and mental health than non-immigrants. Previous studies hypothesized that low vitD concentration was associated with depression [48,49]. Another study found that anxiety, depression, and health-related quality of life were not associated with S-25(OH)D levels among the immigrant population [50]. Our analysis revealed that the prevalence of CDs was higher among those with older age, lower household income, and obesity. White ethnic groups had a higher prevalence of CDs than non-white groups, and CDs varied among immigrants from different ethnic groups and countries and regions of birth. Our finding concerning the health advantage of being immigrants (i.e., fewer CDs) was observed among Chinese, Black, Filipino, and Arab ethnicities as well as those born in South/Central America and the Caribbean, Africa, Asia, and the Philippines. In contrast, this effect was less consistent (i.e., more CDs) for those born in Italy. However, the difference in health status among immigrants may reflect variations in acculturation. The effect of origin in terms of health, life expectancy, and mortality patterns among immigrant populations varies across different ethnic groups [51].

There was no association between S-25(OH)D and CDs in our adjusted regression model. However, in addition to the environmental and lifestyle changes after immigration, we assumed the cumulative impact of immigrants’ lower levels of S-25(OH)D and the deviation in CD-related biomarkers over time played a crucial role in the subsequent health deterioration among immigrants in terms of the investigated proxy indicators. Furthermore, we assumed that improving S-25(OH)D over time, as found in our previous analysis [2], did not enhance the deviated biomarkers or cure CDs once they had developed. In a randomized control trial, vitD supplementation for patients with type 2 diabetes and asymptomatic vitD deficiency did not improve HbA1c levels [47].

However, considering the consistency of results related to higher deficiency levels of S-25(OH)D among immigrants and the relatively new findings relating to vitD and health deterioration, we cannot quantify the degree and time of deterioration nor guarantee such an association. Therefore, we recommend further research on this topic using longitudinal designs. Immigrants can be followed over their residency time in the host country to explain the patterns, developments, and direction of health deterioration.

Cross-sectional surveys such as the CHMS are not usually used to examine changes in health over time. The bidirectional association between serum S-25(OH)D and CDs is complex, and it is challenging to address residual confounding using an observational study design. Longitudinal observation and interventional trials are warranted to further understand the relationship between S-25(OH)D and immigrants’ health deterioration.

A limitation of using secondary data is that some of the chronic diseases could not be included in the analysis because the date of diagnosis was not provided. The CHMS data did not differentiate immigrants by specific immigration class or type (e.g., refugees, family class, economic migrants, or asylum seekers). Moreover, native-born Canadians formed a single group, despite some being second- or third-generation immigrants; this resulted in a high level of heterogeneity in the reference population. Previous immigration studies recommended assessing vitD status and its determinants among subgroups living in the same country [52]. To compare the same generation of immigrants rather than aggregated generations [53], it will be necessary to gather evidence and formulate recommendations specific to sub-populations that may differ from the overall immigrant population [32].

## 5. Conclusions

This study confirmed that the current health status of immigrants is good (healthy), but over time they may experience more CDs than their non-immigrant counterparts. S-25(OH)D is associated with environmental and behavioral factors and acts as a biological measure but is not associated with immigrants’ current health status. However, in addition to the adoption of unhealthy diets and lifestyles, lower levels of accumulated S-25(OH)D may impact the health profile of immigrants in terms of CD-related biomarkers. This may partially explain immigrants’ health deterioration over time. Given the emerging research interest in immigrants’ health deterioration, we recommend further longitudinal research to investigate the relationship between vitD and health deterioration, accounting for the dietary and lifestyle acculturation process.

### Summary of Key Issues and Findings 

Extensive research evidence indicates that Canadian immigrants tend to be healthier at the time of immigration than well-established immigrants and non-immigrants. This “healthy immigrant” effect lessens over time for immigrants living in Canada. Nevertheless, immigrants are at higher risk for vitD deficiency compared with non-immigrants, with the risk highest in younger and more recent immigrants. The longer an immigrant lived in Canada, the better their serum vitD levels; ethnicity was also a significant indicator of vitD deficiency.

This study highlighted that immigrants tend to have better health than their counterparts in terms of the prevalence of chronic diseases (CDs) such as heart disease, cancer, asthma, and inflammatory lung diseases, but there is no difference in self-rated general and mental health. In contrast, CD-related biomarkers (e.g., glycated hemoglobin and total cholesterol/high-density lipoprotein cholesterol ratio) and serum vitD differed in favor of non-immigrants. There was no difference in the pre-migration prevalence of CDs compared with post-immigration; however, the longer immigrants lived in Canada, the higher the prevalence of CDs. The adjusted regression model showed serum vitD was not associated with CDs. Our findings suggest the cumulative impact of lower serum vitD among im-migrants over time, changing biomarkers, and environmental and lifestyle changes after immigration may play crucial roles in subsequent health deterioration among immigrants.

Further longitudinal research is needed to clarify immigrants’ health deterioration and the complicated and controversial bidirectional association between serum vitD and CDs. For instance, immigrants may be followed over their residency in their host country to explore the patterns, developments, and direction of health deterioration.

## Figures and Tables

**Table 1 nutrients-14-01760-t001:** Weighted prevalence of chronic diseases by sociodemographic and behavioral characteristics and immigration status.

		No CDs ^†^ (78.3%), %	CDs (21.7%), %	*p*-Value
Immigration status	Non-immigrant ^†^	76.80	23.20	<0.001
	Immigrant	83.36	16.64	
Age at immigration, years	<18 years ^†^	91.69	8.31	<0.001
	≥18	74.75	25.25	
Years after immigration	≤5 years ^†^	91.72	8.28	0.004
	>5	80.98	19.02	
Years after immigration	≤10 years ^†^	92.84	7.16	<0.001
	>10	77.34	22.66	
Sex	Male ^†^	78.67	21.33	0.641
	Female	77.88	22.12	
Age group, years	<5	92.92	7.08	<0.001
	5–11	89.83	10.17	
	12–17	85.14	14.86	
	18–64 ^†^	80.01	19.99	
	>64	52.7	47.3	
Household income (CAD)	<50,000	72.34	27.66	<0.001
	50,000–100,000 ^†^	80.35	19.65	
	>100,000	83.09	16.91	
BMI (kg/m^2^)	Underweight	75.95	24.05	<0.001
	Normal weight ^†^	84.72	15.28	
	Overweight	78.64	21.36	
	Obese	67.56	32.44	
Physical activity	Yes^†^	84.36	15.64	<0.001
	No	74.58	25.42	
Education	>Secondary school ^†^	78.72	21.28	0.660
	≤Secondary school	77.87	22.13	
Smoking status	No/former ^†^	77.74	22.26	0.137
	Current smoker	76.27	23.73	
Alcohol status	No/former ^†^	74.15	25.85	0.155
	Current drinker	77.6	22.4	

^†^ Reference value; BMI, body mass index; CAD, Canadian dollars; *p* ≤ 0.05.

**Table 2 nutrients-14-01760-t002:** Weighted prevalence of chronic diseases, chronic-disease-related biomarkers, and self-rated general and mental health by immigration status.

		**Non-Immigrants ^†^** **(78.1%), %**	**Immigrants** **(21.9%), %**	**All Participants,** **(100%), %**	***p*-Value**
Diabetes	(Yes)	4.82	6.59	5.21	0.095
Heart disease	(Yes)	3.55	2.37	3.29	0.029
Cancer	(Yes)	6.41	4.65	6.02	0.041
Asthma	(Yes)	11.46	5.06	10.05	<0.001
Inflammatory lung diseases ^a^	(Yes)	3.86	1.95	3.34	0.028
CDs ^b^	(Yes)	23.2	16.64	21.72	<0.001
Self-rated general health	(Poor/fair)	9.67	11.67	10.11	0.121
Self-rated mental health	(Poor/fair)	8.31	6.76	7.95	0.334
S-25(OH)D, nmol/L	(<50)	31.75	52.82	36.34	<0.001
**CD-Related Biomarkers**		**Non-Immigrants ^†^** **Mean (SE)**	**Immigrants** **Mean (SE)**	**95% CI**	***p*-Value**
Serum vitD	S-25(OH)D (nmol/L)	62.72 (1.73)	51.23 (1.41)	8.37, 14.62	<0.001
Glucose homeostasis	HbA1c (%)	5.39 (0.03)	5.52 (0.04)	−0.19, −0.07	<0.001
Lipid profile	TC/HDL ratio	3.63 (0.035)	3.79 (0.07)	−0.30, −0.02	0.030
Immune system and inflammation	hs-CRP (mg/L)	2.46 (0.07)	2.12 (0.13)	0.06, 0.6	0.017
	IgE (IU/mL)	118.37 (6.46)	102.75 (6.59)	−1.14, 32.38	0.066
Hematology	Ferritin (µg/L)	103.07 (2.42)	123.07 (6.20)	−34.45, −5.54	0.009
	Hb (g/L)	140.54 (0.43)	137.73 (0.70)	1.22, 4.42	0.001

^a^ Inflammatory lung diseases: bronchitis + chronic obstructive pulmonary disease + emphysema; (age > 29 years); ^b^ CDs (chronic diseases): at least one of the listed CDs (diabetes, heart disease, cancer, asthma, and inflammatory lung diseases). ^†^ Reference value; *p* ≤ 0.05; SE, standard error; CI, confidence interval.

**Table 3 nutrients-14-01760-t003:** Weighted mean 25(OH)D (nmol/L), prevalence of vitD insufficiency (<50 nmol/L), and vitD supplement use by immigration status.

	S-25(OH)D, (nmol/L)				<50 (nmol/L)			vitD Supplement Use		
	Non-Immigrant ^†^ Mean (SE)	Immigrant Mean (SE)	95% CI	*p*-Value	Non-Immigrant ^†^ %	Immigrant %	*p*-Value	Non-Immigrant ^†^ %	Immigrant %	*p*-Value
Diabetes	65.30 (4.05)	47.56 (4.46)	−31.35, −4.13	0.013	31.43	61.63	<0.001	5.28	6.31	0.642
Heart disease	66.63 (2.15)	57.87 (3.80)	−17.42, −0.11	0.048	32.12	38.38	<0.001	^c^	^c^	
Cancer	70.71 (2.86)	67.96 (5.71)	−16.49, 10.98	0.681	29.08	32.28	<0.001	5.02	8.14	0.083
Asthma	59.13 (2.37)	54.23 (3.36)	−12.97, 3.18	0.222	30.68	53.59	<0.001	^c^	^c^	
Inflammatory lung diseases ^a^	63.87 (1.80)	53.27 (5.80)	−22.09, 0.88	0.069	30.76	3841	<0.001	^c^	^c^	
CDs ^b^	62.18 (1.80)	56.29 (2.49)	−11.23, −0.53	0.033	31.35	47.38	<0.001	4.91	4.97	0.379
Self-rated general health (poor/fair)	58.69 (1.94)	48.68 (2.98)	3.61, 16.43	0.004	31.13	61.2	<0.001	5.28	4.37	0.807
Self-rated mental health (poor/fair)	59.83 (2.43)	47.55 (3.75)	4.12, 20.45	0.005	32.52	64.39	<0.001	5.67	3.67	0.271

^a^ Inflammatory lung diseases: bronchitis + chronic obstructive pulmonary disease + emphysema; (age > 29 years). ^b^ CDs (chronic diseases): at least one of the listed CDs (diabetes, heart disease, cancer, asthma, and inflammatory lung diseases). ^c^ Data are not sufficient for analysis. ^†^ Reference value; *p* ≤ 0.05; SE, standard error; CI, confidence interval.

**Table 4 nutrients-14-01760-t004:** Weighted mean 25(OH)D (nmol/L) based on the time of diagnosis (before or after immigration).

	S-25(OH)D, (nmol/L)			
	Before Immigration ^†^ Mean (SE)	After Immigration Mean (SE)	95% CI	*p*-Value
Diabetes	41.80 (3.87)	48.48 (5.31)	−20.59, 7.24	0.331
Heart disease	44.38 (8.01)	59.55 (4.346)	−35.52, 5.18	0.136
Cancer	71.37 (16.93)	67.8 (5.95)	−36.74, 43.80	0.858
Asthma	52.82 (5.88)	56.57 (4.63)	−18.74, 11.25	0.609

^†^ Reference value; *p* ≤ 0.05; SE, standard error; CI, confidence interval.

**Table 5 nutrients-14-01760-t005:** Multivariable logistic regression analysis for chronic diseases and serum 25(OH)D (contentious) in a predefined model.

		CDs		
		OR (SE)	95% CI	*p*-Value
S-25(OH)D, nmol/L	1 nmol/L	1.007 (0.004)	1.00, 1.01	0.070
Immigration status	Immigrants	0.40 (0.09)	0.25, 0.63	<0.001
Age, years	1 year	1.02 (0.008)	1.00, 1.04	0.024
Sex	Female	0.86 (0.18)	0.56, 1.33	0.494
Household income, CAD (reference < 50,000)	50,000–100,000	0.61 (0.11)	0.42, 0.88	0.011
	≥100,000	0.56 (0.15)	0.32, 0.99	0.046
Ethnic group	Non-white	1.00 (0.29)	0.54, 1.86	0.998
BMI, kg/m^2^ (reference normal weight)	Obese	1.92 (0.47)	1.15, 3.20	0.014
Met physical activity recommendations	No	0.80 (0.21)	0.46, 1.38	0.408
Education	≤Secondary school	0.97 (0.19)	0.69, 1.39	0.841
Smoking status	Current smoker	1.45 (0.34)	0.89, 2.36	0.126
Alcohol	Current drinker	0.64 (0.17)	0.37, 1.12	0.114
VitD supplement or analog use	Yes	0.83 (0.25)	0.45, 1.54	0.540
HbA1c (%)	1%	2.63 (0.37)	1.96, 3.51	<0.001
IgE (IU/mL)	1 IU/mL	1.01 (0.00)	1.00, 1.001	0.001
TC/HDL ratio	Ratio	0.98 (0.10)	0.80, 1.21	0.843
hs-CRP mg/L	1 mg/L	1.01 (0.04)	0.92, 1.10	0.889
Ferritin (µg/L)	1 µg/L	1.00 (0.00)	1.00, 1.001	0.380
Hb (g/L)	1 g/L	0.98 (0.01)	0.97, 1.00	0.060
Constant		0.005 (0.01)	0.000, 0.10	0.001

CDs (chronic diseases): at least one of the listed CDs (diabetes, heart disease, cancer, asthma, and inflammatory lung diseases). Adjusted for immigration status, age, sex, household income, education, ethnicity, BMI, physical activity, smoking behavior, alcohol consumption, vitD supplement use, HbA1c (glycated hemoglobin), TC/HDL-R (total cholesterol/high-density lipoprotein cholesterol ratio), Hb (hemoglobin), ferritin, IgE (immunoglobulin E), and hs-CRP (high-sensitivity C-reactive protein).

## Data Availability

The data described in the manuscript, codebook, and analytic codes are not publicly available because the data are confidential national data hosted by Statistics Canada.

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
