# Peer review of "Vitamin D and Chronic Diseases among First-Generation Immigrants: A Large-Scale Study Using Canadian Health Measures Survey (CHMS) Data"

_nutrients, 2022, doi:10.3390/nu14091760_

Round 1
Reviewer 1 Report
This article is of great interest because it makes a comparison of the immigrant and non-immigrant population of a country. Immigration, whether due to globalization, poverty or war, is increasingly common, and it is of great importance to investigate the risks that arise in populations that change their residence from their home country to a host country with different food, culture, and climate.
-I would suggest to check and confirm format in tables 1(green mark) and 4 (crossed-out text).
-Also please check on the format of Table A4 (cannot see the whole table) and A6 (footnote symbol is above the table).
Author Response
Thank you so much for your valuable comments. All were considered and modified as recommended. The responses for each point are cited below:
- I would suggest to check and confirm format in tables 1 (green mark) and 4 (crossed-out text).
Response:
Thank you, the green mark in Table 1 and the crossed-out text in Table 4 were removed, both were typos.
- Also please check on the format of Table A4 (cannot see the whole table) and A6 (footnote symbol is above the table).
Response:
Thank you, Table A4 was reformatted again to appear in landscape format and all columns are now clearly presented. The footnote symbol of Table A6 was corrected.
Thank you,
Reviewer 2 Report
The main finding of the manuscript, that "Vitamin D and chronic diseases among first-generation immi[1]grants: A large-scale study using Canadian Health Measures Survey (CHMS) data" may be an issue in clinical practice but some concerns should be clarified.
- No study flow chart is presented in the manuscript. Please provide the detailed flow chart of this study!
- Osteoporosis is a disease that is characterized by low bone mass with microarchitectural disruption and skeletal fragility, resulting in an increased risk of fracture. In clinical practice, it is particularly relevant for the aging population, where fragility fractures, reduced glomerular filtration rate (GFR), and low bone mineral density (BMD) are more prevalent. There are clinical impacts on Vitamin D insufficiency or deficiency including chronic kidney disease (CKD) and osteoporosis, which should be analysed and discussed in this manuscript!
Author Response
Thank you so much for your valuable comments. The responses for each point are cited below:
- No study flow chart is presented in the manuscript. Please provide the detailed flow chart of this study!
Response:
We assumed that this study does not require a flow chart because the study’s findings were presented based on the total number of participants in cycles 3 and 4 (11579 participants). The only exclusion was the participants who did not have serum 25(OH)D (570 participants) when the result related to the serum and their cut-offs. The CHMS data is national survey data hosted by Statistics Canada under a very restricted confidential policy and most of the details are internally available only to the researchers. Following these restrictions, in our previous publication, we were allowed to publish the total number of participants (included and excluded), the response rate at the national level, and the method of combining the two cycles including the demographic characteristics of each cycle, https://www.mdpi.com/2072-6643/13/8/2702.
- Osteoporosis is a disease that is characterized by low bone mass with microarchitectural disruption and skeletal fragility, resulting in an increased risk of fracture. In clinical practice, it is particularly relevant for the aging population, where fragility fractures, reduced glomerular filtration rate (GFR), and low bone mineral density (BMD) are more prevalent. There are clinical impacts on Vitamin D insufficiency or deficiency including chronic kidney disease (CKD) and osteoporosis, which should be analysed and discussed in this manuscript!
Response:
Thank you, it is a very important point. However, using secondary data is a limitation. The analysis was limited to the non-skeletal chronic diseases that have the date of diagnosis. Kidney disease was part of the data but without the date of diagnosis. BMD and fracture were not part of this data.
Thank you,
Reviewer 3 Report
I had to revise the article entitled ”Vitamin D and chronic diseases among first-generation immigrants: A large-scale study using Canadian Health Measures Survey (CHMS) data” submitted for publishing in Nutrients journal.
The role of vitamin D in development of chronic diseases is an interesting topic that received a large attention in the last decade. The present study highlighted that immigrants tend to have better health than their counter-parts, while CD-related biomarkers and serum vitD differed in favor of non-immigrants. The study suggested that a lower serum vitD among immigrants over time may play crucial roles in subsequent health deterioration among immigrants along with other factors. The research is an interesting, organized study but it requires major revision before publication.
- The first paragraph entitled “The key issues and findings” should be replaced in discussion section.
- Introduction, paragraph 3 – “Moreover, Jablonski et al. (2012)….., please exclude the year.
- The last paragraph of introduction is part of Material and Method. The authors should replace the following sentences and they should maintain only the objectives of the study in the introduction section.
This study used the prevalence of CDs, CD-related biomarkers, and poor/fair self-rated general and mental health as indicators of health status. We hypothesized that higher S-25(OH)D would be protective against CDs (as the main indicator for health de-terioration). Health and vitD status among first-generation immigrants (foreign-born) were ascertained using Canadian Health Measures Survey (CHMS) data and compared with those in the non-immigrant (native-born) population.
- In Material and Method section/Measures the authors explained how they stratified the patients based on vitamin D level in 2 groups: sufficient and insufficient vitamin D. Why they did not stratify in 3 groups: deficient<25 nmol/L, insufficient 25-50 nmol/l and sufficient > 50 nmol/l as most of the guidelines.”
- The authors defined five chronic conditions - diabetes, heart disease, cancer, asthma, and inflammatory lung diseases. The authors should explain why they considered separately asthma from the other inflammatory lung diseases, since asthma is a chronic inflammatory disease. What entities they included in inflammatory lung disease. The authors should also clarify what that means heart disease? They refer to ischemic heart disease or in general to heart diseases.
- Results section/table 1 – the results for reference criteria should be also included, otherwise it is difficult for readers to compare the results.
- Table 2 – the sum of percentages for non-immigrants and immigrants exceed 100%. Please check the original data.
- Table 4 – What represent the following values: 43/32.12, etc
- Table A4 is not completed. I could not analyze it
- The authors should carefully check all the references – there are errors of typing or errors of citation (ref 5 – it is a book or an article? , ref 6 and 7 – I did not find the journals mentioned in the citations, ref 9 and 13 – the website links are missing, ref 16 – the journal or the editor are missing)
Author Response
Thank you so much for your valuable comments. All were reviewed and modified as recommended. The responses for each point are cited below:
- The first paragraph entitled “The key issues and findings” should be replaced in the discussion section.
Response:
Thank you, the section of “The key issues and findings” was moved to the discussion section as requested in a box with a title called “Summary of the key issues and findings”.
- Introduction, paragraph 3 – “Moreover, Jablonski et al. (2012)….., please exclude the year.
Response:
The year of “2012” in paragraph 3 was removed.
- The last paragraph of introduction is part of Material and Method. The authors should replace the following sentences and they should maintain only the objectives of the study in the introduction section.
Response:
Thank you, the mentioned text related to the material and methods was removed, and part of the text was moved into the method section, the second paragraph.
- In Material and Method section/Measures the authors explained how they stratified the patients based on vitamin D level in 2 groups: sufficient and insufficient vitamin D. Why they did not stratify in 3 groups: deficient<25 nmol/L, insufficient 25-50 nmol/l and sufficient > 50 nmol/l as most of the guidelines.”
Response:
Thank you, it is a valid concern because these cut-offs are important for clinicians and readers. Not using these cut-offs was for the following reason: the number of patients in each category of CDs will not be sufficient to be stratified into 3 groups and will not fulfil the restricted stratification policy of Statistics Canada in publishing the results. The explanation of why we used the cut-off (<50 vs. ≥50) was given in the method section, first paragraph under Measures.
” We used S-25(OH)D cut-off points to identify vitD status as either sufficient (≥50 nmol/L) or insufficient (<50 nmol/L), as defined by the Institute of Medicine (IOM) and other experts [12, 38, 39].”
- The authors defined five chronic conditions - diabetes, heart disease, cancer, asthma, and inflammatory lung diseases. The authors should explain why they considered separately asthma from the other inflammatory lung diseases, since asthma is a chronic inflammatory disease. What entities they included in inflammatory lung disease. The authors should also clarify what that means heart disease? They refer to ischemic heart disease or in general to heart diseases.
Response:
Thank you.
Emphysema and chronic bronchitis are the two most common conditions that contribute to COPD as a chronic inflammatory lung disease. These two conditions usually occur together and can vary in severity among individuals with COPD. However, asthma is a chronic inflammatory airway disease that may be a risk factor for developing COPD. Though immunological and inflammatory backgrounds are shared in both COPD and asthma, two distinct pathophysiological changes accompany each of the two diseases
https://pubmed.ncbi.nlm.nih.gov/10421834/#:~:text=Abstract,of%20professional%20antigen%2Dpresenting%20cells.
https://www.mayoclinic.org/diseases-conditions/copd/symptoms-causes/syc-20353679
Moreover, the number of patients with asthma was more than 1000, while COPD, emphysema, and chronic bronchitis were only around 230 patients. We assumed that merging these four conditions together will overestimate the prevalence of this condition and may entail inaccurate estimation and give an erroneous conclusion on the relationship with vitD among the Canadian population surveyed.
The term heart disease is a general term used by Statistics Canada to describe a collection of diseases and conditions that affects the structure or the function of the heart and are not limited to ischemic heart disease. A short clarification was added to the method section, the second paragraph under Measures.
- Results section/table 1 – the results for reference criteria should be also included, otherwise it is difficult for readers to compare the results.
Response:
Thank you so much, we added all the results to Table 1, including the reference groups. here is the table as well.
|
|
|
No CDs † (78.3%), % |
CDs (21.7%), % |
P-value |
|
Immigration status |
Non-immigrants† |
76.80 |
23.20 |
<0.001 |
|
|
Immigrant |
83.36 |
16.64 |
|
|
Age at immigration, years |
<18 years† |
91.69 |
8.31 |
<0.001 |
|
|
≥18 |
74.75 |
25.25 |
|
|
Years after immigration |
≤5 years† |
91.72 |
8.28 |
0.004 |
|
|
>5 |
80.98 |
19.02 |
|
|
Years after immigration |
≤10 years† |
92.84 |
7.16 |
<0.001 |
|
|
>10 |
77.34 |
22.66 |
|
|
Sex |
Male† |
78.67 |
21.33 |
0.641 |
|
|
Female |
77.88 |
22.12 |
|
|
Age group, years |
<5 |
92.92 |
7.08 |
<0.001 |
|
|
5–11 |
89.83 |
10.17 |
|
|
|
12–17 |
85.14 |
14.86 |
|
|
|
18–64† |
80.01 |
19.99 |
|
|
|
>64 |
52.7 |
47.3 |
|
|
Household income (CAD) |
<50000 |
72.34 |
27.66 |
<0.001 |
|
|
50000–100000† |
80.35 |
19.65 |
|
|
|
>100000 |
83.09 |
16.91 |
|
|
BMI (kg/m2) |
Underweight |
75.95 |
24.05 |
<0.001 |
|
|
Normal weight† |
84.72 |
15.28 |
|
|
|
Overweight |
78.64 |
21.36 |
|
|
|
Obese |
67.56 |
32.44 |
|
|
Physical activity |
Yes† |
84.36 |
15.64 |
<0.001 |
|
|
No |
74.58 |
25.42 |
|
|
Education |
>Secondary school† |
78.72 |
21.28 |
0.660 |
|
|
≤Secondary school |
77.87 |
22.13 |
|
|
Smoking status |
No/former† |
77.74 |
22.26 |
0.137 |
|
|
Current smoker |
76.27 |
23.73 |
|
|
Alcohol status |
No/former† |
74.15 |
25.85 |
0.155 |
|
|
Current drinker |
77..6 |
22.4 |
|
- Table 2 – the sum of percentages for non-immigrants and immigrants exceed 100%. Please check the original data.
Response:
Thank you so much, corrected as (78.1 vs. 21.9)
- Table 4 – What represent the following values: 43/32.12, etc.
Response:
Thank you, these numbers were removed from table 4 (by mistake presented in this table).
- Table A4 is not completed. I could not analyze it
Response:
Thank you, the page of table A4 was in portrait format and the remaining columns did not appear properly. The page sitting is reformatted again to appear in landscape format and all columns are clearly presented.
- The authors should carefully check all the references – there are errors of typing or errors of citation (ref 5 – it is a book or an article? , ref 6 and 7 – I did not find the journals mentioned in the citations, ref 9 and 13 – the website links are missing, ref 16 – the journal or the editor are missing)
Response:
Thank you so much, all references were reviewed and corrected.
Thank you,
Round 2
Reviewer 2 Report
Thanks for the revision of the manuscript titled “Vitamin D and chronic diseases among first-generation immigrants: A large-scale study using Canadian Health Measures Survey (CHMS) data”. The following is my comment and major concern.
Measures
The S-25(OH)D is expressed in nanomoles per liter (nmol/L) and measured using chemiluminescence immunoassay technology (DiaSorin®, Ltd., Stillwater, MN, USA). The analytical detection limit for S-25(OH)D was 10–375 nmol/L. ……………..
We used the presence of CDs as the outcome-dependent factor for health deterioration. CDs included 24 chronic diseases and conditions (e.g., type 1 and 2 diabetes, heart disease (not limited to ischemic heart disease), and mood disorder).
Please detailed depreciation of these 24 chronic diseases! Is it containing chronic kidney disease (CKD) or other important chronic systemic diseases (such as cardiovascular disease……)? In addition, why authors only chose including Diabetes, Heart disease, Cancer, Asthma, Inflammatory lung diseases in this manuscript? This seems cannot provide adequate information and evidence to the readers about chronic disease!!
Author Response
Thank you for the time and effort in reviewing our manuscript and bringing this issue to our attention. Below we have clarified the issue and have added a limitation on this in the discussion section.
The scope of this study is immigrant health deterioration in relation to vitamin D. In discussions with a clinician who is affiliated with Statistics Canada, a list of the 24 chronic diseases in the CHMS was initially identified (see table below). A final decision was taken to include only the chronic diseases of “non-skeletal health” when the date of diagnosis was given only. More specifically, we did not include all 24 chronic diseases in our analysis for various reasons as follows:
- Some of these chronic diseases were under “skeletal health” which is not in the scope of this study (e.g., osteoporosis).
- Some of these chronic diseases occurred in insufficient numbers that could not be stratified and fulfill Statistics Canada’s confidential policy of publishing results with small numbers, and their exclusion will not greatly impact the results given their limited number (e.g., hepatitis).
- Some of these chronic diseases (e.g., kidney disease) did not have a date of diagnosis, which is essential for the analysis and interpretation that is linked to the date of immigration.
For the chronic diseases that are non-skeletal with a sufficient number of cases that can be published without jeopardizing the confidentiality of the data (e.g., kidney disease), but the date of diagnosis was not available, and it is a limitation. The following statement of limitation has been added to the discussion section:
“A limitation of using secondary data is that some the chronic diseases could not be included in the analysis because the date of diagnosis was not provided.”
|
Diseases and Conditions |
Date of diagnosis |
Skeletal disease |
Insufficient number |
No date of diagnosis |
|
Diabetes (type1 & 2) |
X |
|
|
|
|
Heart disease |
X |
|
|
|
|
Cancer |
X |
|
|
|
|
Asthma |
X |
|
|
|
|
Bronchitis |
X |
|
|
|
|
Emphysema |
X |
|
|
|
|
|
|
|
|
|
|
COPD |
|
|
|
X |
|
Fibromyalgia |
|
X |
|
X |
|
Fibromyalgia and back pain |
|
X |
|
X |
|
Osteoporosis |
|
X |
|
X |
|
Back pain excluding fibromyalgia and arthritis |
|
X |
|
X |
|
Thyroid condition |
|
|
X |
X |
|
High blood pressure |
|
|
|
X |
|
High cholesterol |
|
|
|
X |
|
Liver & gallbladder problems |
|
|
|
X |
|
Kidney disease |
|
|
|
X |
|
Stroke |
|
|
X |
X |
|
Hepatitis |
|
|
X |
X |
|
Mood disorder |
|
|
X |
X |
|
Eating disorder |
|
|
X |
X |
|
Developmental disability |
|
|
X |
X |
|
Attention deficit disorder |
|
|
X |
X |
|
Learning disorders |
|
|
X |
X |
|
Other Phys and Mental Cond |
|
|
|
X |
Reviewer 3 Report
The authors clarified the presentation. I am satisfy with their answers. No further comments.
Author Response
Thank you for the time and effort in reviewing our manuscript. Your valuable comments helped us to clarify and strengthen our manuscript and improve its readability.
Thank you,